# MULTI-MODAL ACTION RECOGNIZER BRIDGES HUMAN MOTION GENERATION AND UNDERSTANDING

## ABSTRACT

Human action recognition and motion generation are two active research problems in human-centric computer vision, both aiming to align motion with textual semantics. However, most existing works study these two problems separately, without uncovering the bidirectional links between them, namely that motion generation requires semantic comprehension. This work investigates unified action recognition and motion generation by leveraging skeleton coordinates for both motion understanding and generation. We propose Coordinates-based Autoregressive Motion Diffusion (CoAMD), which synthesizes motion in a coarse-to-fine manner. As a core component of CoAMD, we design a Multi-modal Action Recognizer (MAR) that provides semantic guidance for motion generation. Our model can be applied to four important tasks, including skeleton-based action recognition, text-to-motion generation, text–motion retrieval, and motion editing. Extensive experiments on 13 benchmarks across these tasks demonstrate that our approach achieves state-of-the-art performance, highlighting its effectiveness and versatility for human motion modeling.

## 1 INTRODUCTION

Human-centric computer vision (Wang et al., 2025b; Tang et al., 2025) seeks to understand and interact with humans through visual data. Its key tasks include person search, pose estimation, action recognition, and attribute recognition. Human motion modeling is a core area of human-centric research. It aims to represent, understand, predict, and generate human movements for animation (Peng et al., 2018), virtual avatars (Peng et al., 2021), behavior analysis (Carreira & Zisserman, 2017), and human–robot interaction (Ravichandar et al., 2020).

Early research in human motion modeling primarily focuses on understanding human actions, particularly skeleton-based action recognition. Representative approaches include hierarchical modeling (Du et al., 2015), which represents actions at multiple levels of spatial granularity; spatio-temporal modeling (Yan et al., 2018), which captures both the spatial configurations and temporal dynamics of the human body; and the two-stream paradigm (Wang & Wang, 2017; Zhu et al., 2023a), which learns representations with separate spatial and temporal streams to enhance action understanding. Although action understanding associates human motion with semantic labels, these semantics are constrained by a limited number of action classes.

With the rapid development of text-to-image generation (Ramesh et al., 2022; Saharia et al., 2022), generating human motion from textual descriptions has gained increasing attention (Zhu et al., 2023b). Text-to-motion generation aims to synthesize realistic and semantically consistent human motions conditioned on natural language, bridging the gap between high-level semantic understanding and low-level motion dynamics. Methods based on diffusion models (Tevet et al., 2023; Zhang et al., 2024a), autoregressive transformers (Pinyoanuntapong et al., 2024a), and large language models (LLMs) (Jiang et al., 2023) have been successfully applied to text-to-motion generation. While existing diffusion-based and autoregressive approaches effectively capture temporal dependencies and generate realistic, smooth motions, they are primarily designed for one-way text-to-motion generation and are less capable of handling motion-to-semantic tasks.

Although skeleton-based action recognition and text-to-motion generation are two important tasks of human motion modeling. Most existing works study the two tasks separately, without exploring their

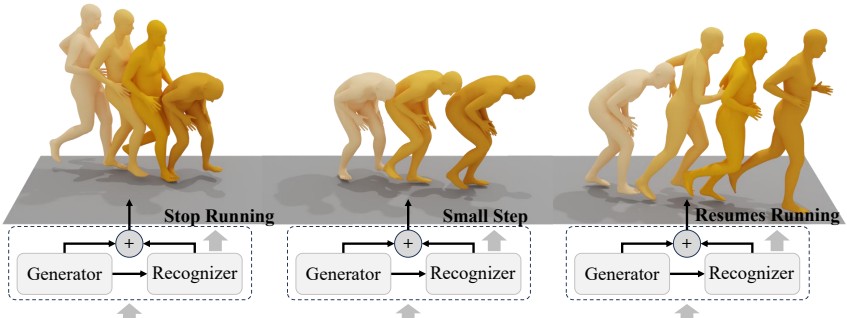

A person who is running, stops, bends over and looks down while taking small steps, then resumes running

Figure 1: Our framework operates in an iterative loop where a Generator synthesizes a portion of the motion, and a Recognizer facilitates its semantic alignment with the text. The recognizer's feedback guides the generator's next step, progressively refining the motion to match complex descriptions.

potential connections. However, these two areas are inherently connected, as they both build alignment between motion dynamics and textual semantics. Action understanding provides structured knowledge of motion dynamics and semantics, which can guide controllable generation; conversely, generative models capture rich motion priors that can enhance recognition through data augmentation and spatio-temporal modeling. Bridging the gap between understanding and generation not only enables a unified framework for bidirectional tasks such as motion-to-text and text-to-motion, but also fosters more robust and generalizable human motion modeling.

In this work, we propose a novel framework that bridges the gap between motion generation and action recognition. Our core idea is to leverage a well-trained Multi-modal Action Recognizer (MAR) not merely as a recognition tool, but as an active semantic guide within the generative process itself, as illustrated in Figure 1. To achieve this, we introduce a multi-modal motion representation that decomposes absolute coordinates into joint, bone, and motion streams, thereby providing a superior foundation for both understanding and synthesis. We then build a motion generator upon Coordinates-based Autoregressive Motion Diffusion (CoAMD), which iteratively synthesizes motion in a coarse-to-fine manner. The MAR is trained with dual objectives of fine-grained retrieval and high-level classification. During the autoregressive generation, at each step, we use the MAR to compute a semantic alignment score for the partially generated motion. The gradient of this score is then used to steer the sampling trajectory of the diffusion model, correcting the motion in real-time to better align with the textual semantics. This tight integration of recognition and generation allows our model to produce motions of exceptional fidelity and semantic accuracy.

Our main contributions can be summarized as follows:

- We introduce a unified framework that integrates human motion generation with skeleton-based action recognition, establishing a bidirectional connection between human motion and language semantics.
- We propose Coordinates-based Autoregressive Motion Diffusion (CoAMD), utilizing skeleton coordinates for motion generation while designing a multi-modal action recognizer to provide gradient-based guidance to the motion diffusion model.
- Our method achieves state-of-the-art performance on benchmarks for skeleton-based action recognition, text-to-motion generation, motion editing, and motion–text retrieval, demonstrating both its effectiveness and versatility.

## 2 RELATED WORK

**Skeleton-Based Action Recognition.** Skeleton-based action recognition aims to understand and classify human actions by modeling the dynamics of body joints over time, which are typically represented by absolute coordinates in 3D space (Duan et al., 2022b; Lee et al., 2023; Liu et al., 2025; Wang et al., 2025a). Recent advances mainly follow two architectures: Graph Convolutional Networks (GCNs) and Transformers. GCN-based approaches gain widespread attention due to the

natural graph structure of skeleton data, where joints serve as nodes and bones as edges. The pioneering work of ST-GCN (Yan et al., 2018) first applies graph convolutions to effectively capture the spatio-temporal features of the human skeleton, which spurs a significant amount of follow-up research in this direction (Chi et al., 2022; Duan et al., 2022a; Huang et al., 2023; Zhou et al., 2024b; Xie et al., 2025). In parallel, Transformer-based approaches enter the field for their exceptional ability to model long-range dependencies within sequences (Xin et al., 2023; Wang & Koniusz, 2023; Wu et al., 2024; Do & Kim, 2024), a crucial aspect for understanding complex, long-term actions. With the growing demand for recognizing unseen actions, research focus increasingly shifts towards Zero-Shot Action Recognition (ZSAR) (Zhou et al., 2023; Kuang et al., 2025; Wu et al., 2025c; Zhu et al., 2025; Chen et al., 2025), which aims to classify action categories by aligning visual and semantic spaces. We build upon the transformer-based paradigm for action recognition and adopt a multi-modal approach to capture complex, long-term actions. By leveraging contrastive learning to align visual and semantic spaces, our method supports open-set recognition tasks and further guides the diffusion model to achieve more effective motion generation.

**Text-to-Motion Generation.** Text-to-motion generation aims to synthesize realistic human motion from natural language descriptions. Some research adopts VQ-based autoregressive approaches, which discretize continuous motion into a sequence of tokens and employ autoregressive models for generation (Guo et al., 2022b; Zhang et al., 2023a; Yuan et al., 2024; Pinyoanuntapong et al., 2024b; Guo et al., 2024; Pinyoanuntapong et al., 2024a; Lu et al., 2025; Zhang et al., 2025b; Cao et al., 2025). Meanwhile, diffusion models emerge as a dominant paradigm for motion generation, inspired by their success in image synthesis Kim et al. (2023); Chen et al. (2023); Yuan et al. (2023); Zhou et al. (2024a); Zhang et al. (2025a); Tevet et al. (2025); Zhao et al. (2025); Jin et al. (2023; 2024); Li et al. (2024a); Liu et al. (2024). Pioneering works such as MDM and MotionDiffuse demonstrate that progressively denoising a Gaussian distribution under text conditions produces diverse and high-quality motions (Tevet et al., 2023; Zhang et al., 2024a). A cornerstone of modern text-to-motion generation is the HumanML3D representation (Guo et al., 2022a). By encoding motion using relative coordinates and incorporating built-in redundant features like local velocities and relative rotations, it significantly simplifies the learning task and becomes the mainstream choice for subsequent methods. Later, the MARDM analyzes and simplifies these redundant features to better suit diffusion models (Meng et al., 2025b). Recently, ACMDM challenges this paradigm by achieving state-of-the-art results using simple absolute joint coordinates (Meng et al., 2025a). Building on these insights, we adopt absolute coordinates but advances this representation through a multi-modal perspective within the diffusion-based autoregressive framework. We argue that this decomposed, multi-modal representation of absolute coordinates not only facilitates the generation of higher-fidelity motion but also builds a natural bridge to the task of action recognition.

**Unified Human Motion Generation and Understanding.** Instead of studying them separately, recent works aim to unify human motion generation and understanding. LaMP (Li et al., 2025b) introduces a language–motion pretraining framework, improving alignment for text-to-motion generation, motion–text retrieval, and motion captioning. KinMo (Zhang et al., 2024c) decomposes motion into joint-group movements and interactions, using hierarchical semantics and coarse-to-fine synthesis to enable fine-grained text-to-motion retrieval, generation and precise joint control. LMM (Zhang et al., 2024b) unifies diverse motion generation tasks through consolidated datasets, articulated attention, and large-scale pre-training. MG-MotionLLM (Wu et al., 2025a) handles fine-grained motion generation with understanding across multiple temporal granularities. UniMotion (Li et al., 2025a) jointly supports flexible motion control and frame-level motion understanding. Lyu et al. (2025) introduce a lexicalized, sparse motion–language representation to produce interpretable motion descriptors and improve cross-modal alignment. Motion-Agent (Wu et al., 2025b) encodes motions into discrete tokens aligned with LLM vocabularies and enables multi-turn interactive generation. DRL (Liang et al., 2025) introduces a closed-loop reciprocal learning framework where motion understanding and generation supervise each other. However, these works treat motion understanding merely as motion–text retrieval and motion captioning, and no existing study has yet bridged skeleton-based human action recognition with motion generation, a gap our work seeks to fill.

## 3 METHODOLOGY

Our goal is to develop a unified human motion modeling framework that not only produces controllable and semantically accurate motions but also enables a deeper understanding of

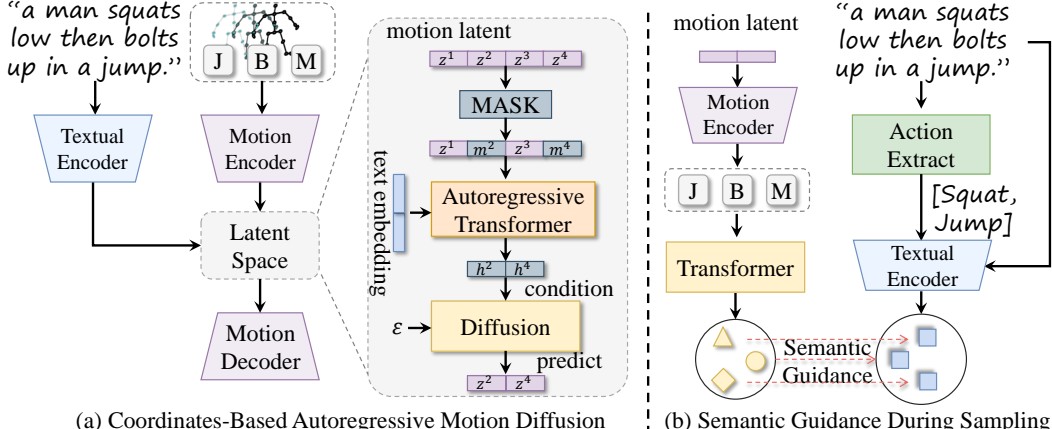

(a) Coordinates-Based Autoregressive Motion Diffusion    (b) Semantic Guidance During Sampling

Figure 2: An overview of the CoAMD architecture. (a) The main generative model, which uses a Motion Encoder-Decoder (AE) to map multi-modal inputs into a latent space. A Masked Autoregressive Transformer then processes this latent sequence, with a Diffusion model responsible for filling in the masked tokens. (b) The semantic guidance mechanism during sampling. The MAR computes a semantic alignment score from the decoded motion, and the gradient of this score is used to refine the latent prediction from the diffusion model, ensuring better alignment with the text.

the action semantics of human motion. To this end, we propose a novel framework that synergizes a multi-modal action recognizer, a coordinates-based autoregressive motion diffusion, and a semantic mechanism, with an overview of the architecture depicted in Figure 2.

## 3.1 MULTI-MODAL ACTION RECOGNIZER

The Multi-modal Action Recognizer (MAR), whose architecture is detailed in Figure 3, is a model trained to comprehend action semantics from two complementary perspectives: fine-grained retrieval and high-level recognition. To facilitate this, we first establish multi-label action classification benchmarks on the HumanML3D. We process the original annotations by using an LLM to extract core action verbs, followed by applying a Balanced K-Means clustering algorithm. This consolidates semantically redundant tags (e.g., "walk", "walks", "walking") into a coherent set of action classes.

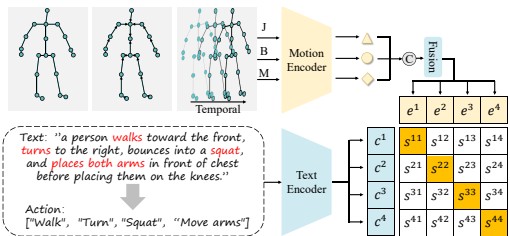

Figure 3: Our Multi-modal Action Recognizer (MAR), trained with a unified contrastive loss to align embeddings and assess both fine-grained and high-level semantic similarity.

The MAR model, $R_\phi$, takes our multi-modal motion representation $(\mathbf{X_j}, \mathbf{X_b}, \mathbf{X_m})$ as input and is optimized to produce semantically rich embeddings. The training is driven by a unified contrastive learning objective. Given a batch of $N$ motion-text pairs, we construct sets of motion embeddings $\{\mathbf{e}_i\}_{i=1}^N$ and corresponding text embeddings $\{\mathbf{c}_i\}_{i=1}^N$. The motion embedding set $\mathbf{e}$ includes both the fused representation and the individual modality representations $(\mathbf{e_f}, \mathbf{e_j}, \mathbf{e_b}, \mathbf{e_m})$, while the text embedding set $\mathbf{c}$ includes embeddings for both fine-grained descriptions and high-level action class labels. The model is trained by minimizing the InfoNCE loss across these pairs:

$$\mathcal{L}_{\text{MAR}} = -\frac{1}{N} \sum_{i=1}^N \log \frac{\exp(\text{sim}(\mathbf{e}_i, \mathbf{c}_i)/\tau)}{\sum_{k=1}^N \exp(\text{sim}(\mathbf{e}_i, \mathbf{c}_k)/\tau)}, \tag{1}$$

where $\text{sim}(\cdot, \cdot)$ denotes cosine similarity and $\tau$ is a temperature hyperparameter.

### 3.2 COORDINATES-BASED AUTOREGRESSIVE MOTION DIFFUSION

**Multi-Modal Motion Representation.** Building upon the recent success of using absolute coordinates (Meng et al., 2025a), we introduce a richer, more structured multi-modal motion representation to provide a superior foundation for our model. A given motion sequence, represented by absolute joint coordinates $\mathbf{X_j} \in \mathbb{R}^{L \times J \times 3}$, is decomposed into three complementary streams: the original joints ($\mathbf{X_j}$), bone vectors ($\mathbf{X_b}$), and motion dynamics ($\mathbf{X_m}$). The bone stream, $\mathbf{X_b}$, explicitly models the body's kinematic structure by computing the difference between connected joints. The motion stream, $\mathbf{X_m}$, captures temporal dynamics by calculating the frame-to-frame joint displacement.

These three streams are processed by a Multi-Modal Autoencoder (AE). The AE features shallow, independent projection layers for each modality, followed by an early fusion step and a shared deep encoder $\mathcal{E}$. This maps the multi-modal input to a compact latent representation $\mathbf{z} = \mathcal{E}(\mathbf{X_j}, \mathbf{X_b}, \mathbf{X_m}) \in \mathbb{R}^{L' \times D}$, where $L'$ is the downsampled length and $D$ is the latent dimension. The AE is trained with a multi-modal reconstruction loss. Given the decoded joint sequence $\hat{\mathbf{X}}_\mathbf{j} = \mathcal{D}(\mathbf{z})$, we re-derive the corresponding bone $\hat{\mathbf{X}}_\mathbf{b}$ and motion $\hat{\mathbf{X}}_\mathbf{m}$ streams. The total loss is the sum of the Smooth L1 losses for each stream:

$$\mathcal{L}_{\text{AE}} = \sum_{k \in \{j, b, m\}} \|\mathbf{X}_k - \hat{\mathbf{X}}_k\|_1, \tag{2}$$

The latent representation $\mathbf{z}$ generated under multi-loss constraints exhibits both richness and stability, providing a critical foundation for constructing this diffusion model.

**Autoregressive Human Motion Model.** We adopt the masked autoregressive generation framework (Li et al., 2024b), which has proven highly effective for motion synthesis. This approach simplifies the generation of a full latent sequence $\mathbf{z} \in \mathbb{R}^{L' \times D}$ by iteratively predicting a randomly masked subset of its tokens, conditioned on the unmasked remainder and a text prompt $c$. The architecture consists of two core components: a Masked Autoregressive Transformer $\mathcal{T}$ and a Diffusion Model $v_\theta$. First, the latent sequence $\mathbf{z}$ is partitioned into unmasked tokens $\mathbf{z}_{\text{um}}$ and masked tokens $\mathbf{z}_\text{m}$. The MAR-Transformer $\mathcal{T}$ processes the unmasked sequence $\mathbf{z}_{\text{um}}$ to produce a contextual embedding $\mathbf{h}$ specifically for the masked positions:

$$\mathbf{h} = \mathcal{T}(\mathbf{z}_{\text{um}}, c). \tag{3}$$

This context $\mathbf{h}$ encapsulates all necessary information from the visible parts of the sequence and the text prompt to guide the generation of the missing parts. Next, the generative step for the masked tokens $\mathbf{z}_\text{m}$ is handled by the diffusion model $v_\theta$, which operates on these tokens exclusively. We employ a velocity prediction objective within a flow-matching formulation. The forward process defines a noisy version of the clean masked tokens $\mathbf{z}_{\text{m},0}$ at a continuous timestep $t \in [0, 1)$:

$$\mathbf{z}_{\text{m},t} = (1 - t)\mathbf{z}_{\text{m},0} + t\boldsymbol{\epsilon}, \quad \boldsymbol{\epsilon} \sim \mathcal{N}(0, \mathbf{I}). \tag{4}$$

The diffusion model $v_\theta$ is then trained to predict the velocity required to denoise $\mathbf{z}_{\text{m},t}$, conditioned on the context $\mathbf{h}$ provided by the transformer. The model is optimized by minimizing the difference between its predicted velocity and the ground-truth velocity derived from the clean target tokens $\mathbf{z}_{\text{m},0}$ and the noise $\boldsymbol{\epsilon}$:

$$\mathcal{L}_{\text{diff}} = \mathbb{E}_{\mathbf{z}_{\text{m},0}, \boldsymbol{\epsilon}, t, \mathbf{h}} \|\mathbf{v}_\theta(\mathbf{h}, t) - (\boldsymbol{\epsilon} - \mathbf{z}_{\text{m},0})\|^2. \tag{5}$$

During inference, for each generative step, we first compute the context $\mathbf{h}$ from the currently unmasked tokens, and then use an ODE solver with the learned velocity field $v_\theta$ to sample and fill in the masked tokens.

### 3.3 SEMANTIC GUIDANCE DURING SAMPLING

During the masked autoregressive inference, at each generative step, we leverage the MAR to guide the sampling process. Let $\mathbf{z}$ be the current latent sequence, which contains both previously generated tokens and masked tokens to be filled. The diffusion model first predicts a preliminary update for the masked tokens, resulting in a complete latent sequence $\mathbf{z}'$. We then compute a guidance gradient to refine this prediction.

Table 1: Quantitative comparison for text-to-motion generation on the HumanML3D dataset. All models are assessed using absolute coordinates. We conduct 20 evaluation runs for each metric and report the mean with 95% confidence interval. Bold and underlined values denote the best and second-best results, respectively.

| Methods | R-Precision↑ | | | FID↓ | MM-Dist↓ | MModality↑ | CLIP-score↑ |
|---|---|---|---|---|---|---|---|
| | Top 1 | Top 2 | Top 3 | | | | |
| Real | $0.501^{\pm.002}$ | $0.691^{\pm.002}$ | $0.785^{\pm.001}$ | $0.000^{\pm.000}$ | $3.074^{\pm.005}$ | – | $0.665^{\pm.001}$ |
| MDM-50Step (2023) | $0.435^{\pm.013}$ | $0.627^{\pm.014}$ | $0.737^{\pm.011}$ | $0.395^{\pm.065}$ | $3.443^{\pm.060}$ | $2.182^{\pm.055}$ | $0.578^{\pm.003}$ |
| MotionDiffuse (2024a) | $0.435^{\pm.008}$ | $0.618^{\pm.006}$ | $0.725^{\pm.005}$ | $1.334^{\pm.035}$ | $3.542^{\pm.015}$ | $1.833^{\pm.078}$ | $0.606^{\pm.004}$ |
| ReMoDiffuse (2023b) | $0.467^{\pm.001}$ | $0.654^{\pm.004}$ | $0.748^{\pm.002}$ | $0.207^{\pm.006}$ | $3.289^{\pm.022}$ | $\mathbf{2.560}^{\pm.289}$ | $0.621^{\pm.003}$ |
| MotionLCM V2 (2024) | $0.511^{\pm.007}$ | $\underline{0.707}^{\pm.003}$ | $\underline{0.802}^{\pm.002}$ | $0.152^{\pm.007}$ | $3.005^{\pm.009}$ | $1.993^{\pm.085}$ | $0.640^{\pm.003}$ |
| BAMM (2024a) | $0.455^{\pm.003}$ | $0.645^{\pm.002}$ | $0.746^{\pm.002}$ | $0.310^{\pm.007}$ | $3.348^{\pm.008}$ | $1.950^{\pm.067}$ | $0.614^{\pm.002}$ |
| MoMask (2024) | $0.493^{\pm.002}$ | $0.686^{\pm.002}$ | $0.784^{\pm.002}$ | $\mathbf{0.047}^{\pm.003}$ | $3.124^{\pm.008}$ | $1.356^{\pm.042}$ | $\underline{0.668}^{\pm.001}$ |
| StableMoFusion (2024) | $\underline{0.514}^{\pm.002}$ | $0.706^{\pm.002}$ | $0.801^{\pm.003}$ | $0.111^{\pm.006}$ | $3.023^{\pm.005}$ | $1.811^{\pm.052}$ | $0.666^{\pm.001}$ |
| MARDM-DDPM (2025b) | $0.468^{\pm.003}$ | $0.663^{\pm.003}$ | $0.768^{\pm.002}$ | $0.238^{\pm.008}$ | $3.229^{\pm.010}$ | $\underline{2.438}^{\pm.099}$ | $0.635^{\pm.003}$ |
| MARDM-SiT (2025b) | $0.486^{\pm.003}$ | $0.680^{\pm.003}$ | $0.780^{\pm.002}$ | $0.156^{\pm.007}$ | $3.136^{\pm.010}$ | $2.353^{\pm.101}$ | $0.637^{\pm.002}$ |
| ACMDM-S-PS22 (2025a) | $0.509^{\pm.002}$ | $0.699^{\pm.002}$ | $0.793^{\pm.003}$ | $0.564^{\pm.014}$ | $3.063^{\pm.010}$ | $2.069^{\pm.094}$ | $0.642^{\pm.001}$ |
| CoAMD w/o MAR (ours) | $0.512^{\pm.002}$ | $0.705^{\pm.002}$ | $0.801^{\pm.002}$ | $0.074^{\pm.004}$ | $\underline{2.980}^{\pm.008}$ | $2.012^{\pm.034}$ | $\underline{0.668}^{\pm.001}$ |
| CoAMD (ours) | $\mathbf{0.519}^{\pm.003}$ | $\mathbf{0.708}^{\pm.002}$ | $\mathbf{0.803}^{\pm.002}$ | $\underline{0.065}^{\pm.004}$ | $\mathbf{2.959}^{\pm.009}$ | $2.014^{\pm.036}$ | $\mathbf{0.674}^{\pm.002}$ |

First, the complete latent sequence $\mathbf{z}'$ is decoded into an estimated motion $\hat{\mathbf{X}} = \mathcal{D}(\mathbf{z}')$. Then, a composite semantic alignment score $\mathcal{S}$ is calculated. This score is based on the alignment of the fused and modality-specific embeddings with the target text embedding $c$:

$$\mathcal{S}(\hat{\mathbf{X}}, c) = \sum_{k \in \{f,j,b,m\}} w_k \cdot \text{sim}(R_\phi^k(\hat{\mathbf{X}}), c), \tag{6}$$

where $R_\phi^k$ denotes the specific embedding from the MAR (with $f$ for fused) and $w_k$ are weighting coefficients. With this score, we compute its gradient with respect to the latent sequence $\mathbf{z}'$. To ensure stable updates, the gradient is normalized:

$$\mathbf{g} = \nabla_{\mathbf{z}'} \mathcal{S}(\mathcal{D}(\mathbf{z}'), c), \quad \hat{\mathbf{g}} = \frac{\mathbf{g}}{\|\mathbf{g}\|_2 + \epsilon}. \tag{7}$$

This normalized gradient $\hat{\mathbf{g}}$ points in the direction of steepest ascent for the text-motion alignment score. The final updated latent sequence $\mathbf{z}_{\text{new}}$ is obtained by applying this gradient step only to the tokens that are masked in the current iteration ($M_t$):

$$\mathbf{z}_{\text{new}} = \mathbf{z}' + \gamma \cdot \hat{\mathbf{g}} \odot M_t, \tag{8}$$

where $\gamma$ is a guidance scale hyperparameter and $\odot$ denotes element-wise multiplication. By incorporating this gradient-based guidance step within each iteration of the autoregressive process, we steer the generation towards states that are not only probable under the diffusion model but also well aligned with the detailed semantics of the text prompt.

## 4 EXPERIMENTS

### 4.1 MAIN RESULTS

**Text-to-Motion Generation.** Table 1 and Table 2 present a quantitative comparison of our method CoAMD against state-of-the-art baselines on the HumanML3D and KIT datasets, respectively. Our unguided model (CoAMD w/o MAR), which leverages the proposed multi-modal representation and already demonstrates highly competitive results outperforming most existing methods. This validates the effectiveness of our foundational architecture. When augmented with our Multi-modal Action Recognizer for guidance (CoAMD (ours)), the model's performance is substantially boosted, setting new state-of-the-art scores across nearly all metrics on both benchmarks. The notable improvements in FID and CLIP-Score highlight our guidance mechanism's dual benefit. It enhances the distributional realism of generated motions and simultaneously improves the fine-grained semantic alignment with input text prompts. This demonstrates that integrating an explicit action understanding module into the generative process is a highly effective strategy for advancing text-to-motion synthesis.

Table 2: Quantitative comparison for text-to-motion generation performance on the KIT dataset.

| Methods | R-Precision↑ | | | FID↓ | MM-Dist↓ | MModality↑ | CLIP-score↑ |
|---|---|---|---|---|---|---|---|
| | Top 1 | Top 2 | Top 3 | | | | |
| MDM-50Step (2023) | $0.333^{\pm.012}$ | $0.561^{\pm.009}$ | $0.689^{\pm.009}$ | $0.585^{\pm.043}$ | $4.002^{\pm.033}$ | $1.681^{\pm.107}$ | $0.605^{\pm.007}$ |
| MotionDiffuse (2024a) | $0.344^{\pm.009}$ | $0.536^{\pm.007}$ | $0.658^{\pm.007}$ | $3.845^{\pm.087}$ | $4.167^{\pm.054}$ | $1.774^{\pm.217}$ | $0.626^{\pm.006}$ |
| ReMoDiffuse (2023b) | $0.356^{\pm.004}$ | $0.572^{\pm.007}$ | $0.706^{\pm.009}$ | $1.725^{\pm.053}$ | $3.735^{\pm.036}$ | $\mathbf{1.928}^{\pm.127}$ | $0.665^{\pm.005}$ |
| MoMask (2024) | $0.392^{\pm.006}$ | $0.604^{\pm.008}$ | $0.732^{\pm.006}$ | $0.523^{\pm.022}$ | $3.383^{\pm.030}$ | $\underline{1.892}^{\pm.085}$ | $0.695^{\pm.002}$ |
| MARDM-DDPM (2025b) | $0.375^{\pm.006}$ | $0.597^{\pm.008}$ | $0.739^{\pm.006}$ | $0.340^{\pm.020}$ | $3.489^{\pm.018}$ | $1.479^{\pm.078}$ | $0.681^{\pm.003}$ |
| MARDM-SiT (2025b) | $0.387^{\pm.006}$ | $0.610^{\pm.006}$ | $0.749^{\pm.006}$ | $0.242^{\pm.014}$ | $3.374^{\pm.019}$ | $1.312^{\pm.053}$ | $0.692^{\pm.002}$ |
| ACMDM-S-PS22 (2025a) | $\underline{0.391}^{\pm.005}$ | $\underline{0.615}^{\pm.005}$ | $\underline{0.752}^{\pm.006}$ | $\underline{0.237}^{\pm.010}$ | $\underline{3.368}^{\pm.019}$ | $1.267^{\pm.063}$ | $\underline{0.696}^{\pm.002}$ |
| CoAMD (ours) | $\mathbf{0.431}^{\pm.005}$ | $\mathbf{0.659}^{\pm.007}$ | $\mathbf{0.785}^{\pm.006}$ | $\mathbf{0.217}^{\pm.009}$ | $\mathbf{2.966}^{\pm.019}$ | $1.383^{\pm.067}$ | $\mathbf{0.708}^{\pm.002}$ |

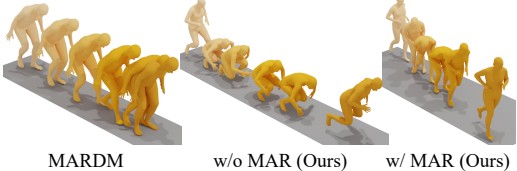 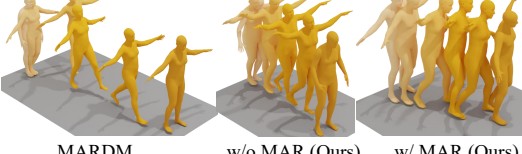

MARDM  w/o MAR (Ours)  w/ MAR (Ours)          MARDM  w/o MAR (Ours)  w/ MAR (Ours)

(a) A person who is running, stops, bends over and looks down while taking small steps, then resumes running.

(b) He stretches both arms out straight to the sides, moves forward with small slow steps, and tries to keep balance while walking.

Figure 4: Qualitative comparison on the HumanML3D dataset. Our guided model (w/ MAR) demonstrates a superior ability to synthesize complex, multi-part action sequences (a) and nuanced stylistic details (b) compared to both the baseline (MARDM) and our unguided version (w/o MAR). Key phrases from the text prompts are highlighted in red.

We provide qualitative comparisons in Figure 4, which further showcase our guided model's superior ability to synthesize complex, multi-part actions and nuanced stylistic details.

**Skeleton-Based Action Recognition.** Since our framework synergizes motion generation and action recognition, we evaluate the effectiveness of the MAR, which provides semantic guidance signals. Using absolute joint coordinates as the foundation of our representation naturally suits the action recognition task. As described in Section 3.1, we establish new action recognition benchmarks on the HumanML3D dataset originally designed for synthesis. Table 3 compares our method against alignment-based action recognition baselines. To ensure fairness, all baseline methods are re-implemented to employ the same visual backbone as our model. The results clearly demonstrate the superiority of our approach. When multi-modal representations are employed (J+B+M), the performance is further and significantly improved. Notably, the most substantial gains are observed in the few-shot classes of both datasets. This strongly supports that explicitly modeling body kinematics (bones) and temporal dynamics (motion) provides richer and more discriminative features. These results validate MAR as a reliable source of semantic guidance for our generation task.

**Cross-Dataset Generalization.** To assess the robustness and transferability of the features learned by our MAR, we evaluate its cross-dataset generalization capability on the NTU-60 and NTU-120 benchmarks. Our MAR is pre-trained solely on the text-motion alignment task of HumanML3D. Following a standard linear probing protocol, we freeze the MAR backbone and train only a linear classification head on the target NTU datasets. To handle heterogeneous skeletons, we map the joints from NTU to their corresponding locations in our model's topology. As shown in Table 4, our

Table 3: Comparison of action recognition performance on the HumanML3D dataset.

| Method | Modality | HumanML3D(%) | | | |
|---|---|---|---|---|---|
| | | Overall | Many-shot | Medium-shot | Few-shot |
| CADA-VAE (Schonfeld et al., 2019) | J | 43.62 | 55.97 | 28.54 | 12.32 |
| SMIE (Zhou et al., 2023) | J | 49.23 | 60.44 | 36.57 | 23.61 |
| DVTA (Kuang et al., 2025) | J | 50.05 | 61.12 | 37.24 | 24.52 |
| CoAMD (ours) | J | 49.11 | 60.13 | 36.14 | 22.91 |
| CoAMD (ours) | J+B+M | **51.63** | **62.24** | **38.13** | **25.32** |

Table 4: Cross-dataset generalization performance on NTU-60 and NTU-120. Our model is pre-trained on HumanML3D and evaluated using a linear probing protocol.

| Method | Modality | NTU-60 | | NTU-120 | |
|---|---|---|---|---|---|
| | | x-sub | x-view | x-sub | x-set |
| UmURL (Sun et al., 2023) | J | 42.31 | 44.98 | 32.14 | 33.81 |
| USDRL (Wang et al., 2025a) | J | 43.48 | 46.23 | 33.02 | 34.69 |
| CoAMD (ours) | J | 45.12 | 48.29 | 34.41 | 36.09 |
| UmURL (Sun et al., 2023) | J+B+M | 47.11 | 50.62 | 36.19 | 37.59 |
| USDRL (Wang et al., 2025a) | J+B+M | 48.02 | 52.08 | 37.01 | 37.98 |
| CoAMD (ours) | J+B+M | **50.24** | **54.04** | **38.29** | **39.11** |

Table 5: Results of motion-text retrieval on the HumanML3D dataset.

| Methods | Text-Motion(%) | | | | | Motion-Text(%) | | | | |
|---|---|---|---|---|---|---|---|---|---|---|
| | R@1 | R@2 | R@3 | R@5 | R@10 | R@1 | R@2 | R@3 | R@5 | R@10 |
| TEMOS (Petrovich et al., 2022) | 40.49 | 53.52 | 61.14 | 70.96 | 84.15 | 39.96 | 53.49 | 61.79 | 72.40 | 85.89 |
| T2M (Guo et al., 2022a) | 52.48 | 71.05 | 80.65 | 89.66 | 96.58 | 52.00 | 71.21 | 81.11 | 89.87 | 96.78 |
| TMR (Petrovich et al., 2023) | 67.16 | 81.32 | 86.81 | 91.43 | 95.36 | 67.97 | 81.20 | 86.35 | 91.70 | 95.27 |
| LaMP (Li et al., 2024c) | 67.18 | 81.90 | 87.04 | 92.00 | 95.73 | 68.02 | 82.10 | 87.50 | 92.20 | 96.90 |
| CoAMD (ours) | **68.33** | **82.61** | **88.05** | **92.45** | **95.97** | **68.86** | **82.77** | **88.09** | **92.36** | **97.21** |

full multi-modal model consistently outperforms baselines across every setting. This performance is noteworthy as the features are learned without exposure to the target datasets or their labels. It validates that our representation captures a transferable understanding of human motion semantics, robust in generalizing to new datasets with different skeletal structures and action vocabularies.

**Text-Motion Retrieval.** Beyond its primary role as a source of semantic guidance for generation, the MAR is an inherently powerful text-motion alignment model capable of cross-modal retrieval. To validate this capability, we evaluate its performance on both text-to-motion and motion-to-text retrieval using the HumanML3D benchmark. As shown in Table 5, we compare our MAR against several state-of-the-art methods designed specifically for motion retrieval. The evaluation is conducted using a standard batch size of 32 for a fair comparison. Our model demonstrates superior performance, achieving the highest scores across nearly all settings. This retrieval capability is a direct result of our dual-task pre-training objective, which forces the model to learn a fine-grained, shared embedding space where motion and text are tightly aligned. This result showcases the versatility of our MAR and provides further evidence for its suitability as a high-quality guidance source.

**Motion Editing.** A key strength of our masked autoregressive framework is its ability to handle motion editing tasks without task-specific modifications. We evaluate on HumanML3D under four standard benchmarks: Temporal Inpainting, Temporal Outpainting, Prefix and Suffix generation. As shown in Table 6, our method outperforms strong baselines across all settings. It achieves higher R-Precision indicating better semantic alignment. It also obtains lower FID and MM-Dist showing improved realism and faithfulness to the ground truth. These results confirm our multi-modal representation and semantic guidance provide a stronger conditioning signal for controllable synthesis.

Table 6: Quantitative comparison on temporal editing tasks on the HumanML3D dataset.

| Tasks | Methods | R-Precision↑ | | | FID↓ | MM-Dist↓ | CLIP-score↑ |
|---|---|---|---|---|---|---|---|
| | | Top 1 | Top 2 | Top 3 | | | |
| Temporal Inpainting | BAMM (Pinyoanuntapong et al., 2024a) | 0.387 | 0.554 | 0.649 | 0.385 | 4.046 | 0.574 |
| | MARDM (2025b) | 0.503 | 0.702 | 0.795 | 0.120 | 3.051 | 0.671 |
| | CoAMD (ours) | **0.518** | **0.716** | **0.804** | **0.023** | **2.943** | **0.677** |
| Temporal Outpainting | BAMM (Pinyoanuntapong et al., 2024a) | 0.433 | 0.605 | 0.707 | 0.206 | 3.615 | 0.613 |
| | MARDM (Meng et al., 2025b) | 0.512 | **0.705** | 0.797 | 0.114 | 3.065 | 0.671 |
| | CoAMD (ours) | **0.518** | 0.703 | **0.800** | **0.022** | **2.967** | **0.675** |
| Prefix | BAMM (Pinyoanuntapong et al., 2024a) | 0.352 | 0.526 | 0.632 | 0.578 | 4.178 | 0.565 |
| | MARDM (Meng et al., 2025b) | 0.515 | **0.709** | 0.800 | 0.120 | 3.039 | 0.673 |
| | CoAMD (ours) | **0.525** | 0.707 | **0.802** | **0.032** | **2.934** | **0.679** |
| Suffix | BAMM (Pinyoanuntapong et al., 2024a) | 0.435 | 0.608 | 0.708 | 0.201 | 3.504 | 0.625 |
| | MARDM (Meng et al., 2025b) | 0.501 | 0.685 | 0.780 | 0.138 | 3.113 | 0.668 |
| | CoAMD (ours) | **0.522** | **0.705** | **0.804** | **0.039** | **3.000** | **0.674** |

Table 7: Impact of different modality combinations on the HumanML3D dataset. Adding Bone (B) and Motion (M) modalities to the baseline Joint (J) representation improves performance on both generation (R-Precision, FID, MM-Dist, CLIP-score) and recognition (Accuracy) tasks.

| Modality | R-Precision↑ | | | FID↓ | MM-Dist↓ | CLIP-score↑ | Top-1 Accuracy(%) | | | |
|----------|-------|-------|-------|-------|----------|-------------|---------|-----------|-------------|----------|
| | Top 1 | Top 2 | Top 3 | | | | Overall | Many-shot | Medium-shot | Few-shot |
| J | 0.512 | 0.705 | 0.801 | 0.074 | 2.980 | 0.668 | 49.11 | 60.13 | 36.14 | 22.91 |
| J+B | 0.515 | 0.706 | 0.802 | 0.072 | 2.975 | 0.670 | 49.85 | 61.02 | 37.00 | 23.65 |
| J+M | 0.517 | 0.707 | 0.803 | 0.069 | 2.970 | 0.672 | 50.22 | 61.53 | 37.52 | 24.10 |
| J+B+M | **0.519** | **0.708** | **0.803** | **0.065** | **2.959** | **0.674** | **51.63** | **62.24** | **38.13** | **25.32** |

Table 8: Ablation study on the MAR with semantic guidance mechanism during sampling.

| method | R-Precision↑ | | | FID↓ | MM-Dist↓ | MModality↑ | CLIP-score↑ |
|--------|-------|-------|-------|-------|----------|------------|-------------|
| | Top 1 | Top 2 | Top 3 | | | | |
| w/o Autoregression | 0.488 | 0.685 | 0.781 | 0.142 | 3.125 | 2.421 | 0.663 |
| w/o MAR | 0.512 | 0.705 | 0.801 | 0.074 | 2.980 | 2.012 | 0.668 |
| w/ MAR(J) | 0.514 | 0.707 | 0.796 | 0.095 | 2.984 | 2.192 | 0.672 |
| w/ MAR(J+B+M) | **0.519** | **0.708** | **0.803** | **0.065** | **2.959** | **2.014** | **0.674** |

## 4.2 ABLATION STUDY

**Effectiveness of Multi-Modal Representation.** We first investigate the impact of our multi-modal representation on both generation and recognition. As detailed in Table 7, we progressively integrate the bone (B) and motion (M) modalities with the baseline joint (J) representation. The results clearly indicate that enriching the representation yields consistent improvements across both tasks. For motion generation, incorporating more modalities enhances the quality of the learned latent space, leading to a steady decrease in the FID score and signifying more realistic motions. For recognition, the additional modalities provide richer features, boosting the MAR's capability and increasing the overall Top-1 Accuracy. This enhanced recognition power provides a more precise guidance signal, raising the R-Precision. This synergistic relationship validates our hypothesis that a unified, multi-modal representation is mutually beneficial for understanding and synthesizing human motion.

**Effectiveness Semantic Guidance for Motion Generation.** Table 8 presents an ablation on our core contribution of iterative semantic guidance. The unguided baseline (w/o MAR) exhibits the lowest overall text-alignment scores. Introducing guidance from the MAR (w/ MAR(J+B+M)) substantially improves performance, reducing FID by nearly 50% and boosting R-Precision, which confirms the efficacy of the guidance signal itself. Critically, to isolate the importance of the iterative application, we test a non-autoregressive variant (w/o Autoregression) where guidance is applied only once. This model's performance falls below the unguided baseline. It demonstrates that a single corrective signal applied to a flawed final output is ineffective. In contrast, our method's step-by-step guidance enables continuous, fine-grained refinements throughout the autoregressive process. This iterative steering prevents the accumulation of semantic errors, validating that the key contribution arises from the tight integration of semantic guidance with the iterative sampling process.

## 5 CONCLUSION

In this work, we introduce CoAMD, a unified framework that integrates text-to-motion generation with skeleton-based action recognition. We show that replacing separate training paradigms with an integrated design, where a strong action recognizer serves as an active semantic guide, yields substantial improvements in both motion quality and text alignment. A central insight is that iterative integration of guidance throughout the masked autoregressive sampling process is essential, as it enables continuous and fine-grained corrections that outperform single post-hoc refinements. Moreover, our multi-modal representation proves mutually beneficial, enriching feature diversity to enhance both generative fidelity and recognition robustness.

**Limitations and Future Work.** Our framework focuses on text-driven single-person motion. Extending it to multi-agent or multi-modal scenarios is a promising direction. While iterative guidance improves semantic precision, it incurs inference overhead, exploring efficient alternatives could balance accuracy and speed. Additionally, ensuring that all generated motions strictly adhere to physical laws is an important direction for future research.

## ETHICS STATEMENT

This work studies motion generation and understanding, which has wide applications in virtual avatars, animation, human-computer interaction, and robotics. We recognize potential ethical risks, including the misuse of generated motions for creating deceptive content or impersonating individuals. Our research is intended purely for academic purposes. All experiments are conducted on publicly available datasets that do not contain personally identifiable or sensitive information. We encourage responsible use of such technologies with appropriate safeguards to mitigate malicious applications.

## REPRODUCIBILITY STATEMENT

We provide all necessary details to reproduce our experiments, including model architectures, training settings and hyperparameters. Our code and pre-trained models will be released to facilitate reproducibility.

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

APPENDIX

In this appendix, we provide additional materials to complement the main text. Specifically, we include datasets (Appendix A.1), examples of HumanML3D annotations (Appendix A.2), implementation details (Appendix A.3), additional qualitative results (Appendix B.1), user study results (Appendix B.2), and detailed mathematical derivations (Appendix C). These materials aim to facilitate a deeper understanding of our methodology and to support reproducibility of the experiments. More results are available on the project website: `https://anonymous.4open.science/w/CoAMD-26D4/`.

# A DATASETS AND IMPLEMENTATION DETAILS

## A.1 DATASETS AND EVALUATION PROTOCOLS

**Action Recognition.** To construct the reward model, we establish multi-label classification benchmarks on HumanML3D (Guo et al., 2022a) by processing and clustering their annotations into 400 classes. Our analysis reveals a significant long-tail distribution within these classes, where a minority of 'head' classes contain the majority of samples. To systematically assess performance under this imbalance, we partition the classes into three frequency-based subsets: Head (top 10%), Medium (next 30%), and Tail (bottom 60%). Furthermore, to evaluate the cross-dataset generalization of our recognizer, we also test its performance on the widely used NTU-RGB+D 60 (Shahroudy et al., 2016) and NTU-RGB+D 120 (Liu et al., 2019) benchmarks.

**Motion Generation.** We validate our approach on two standard text-to-motion benchmarks: HumanML3D and KIT-ML. Consistent with the state-of-the-art paradigm, all models are trained and evaluated using absolute joint coordinates. To ensure a fair and comprehensive comparison, we adopt the robust evaluation framework from (Meng et al., 2025a). Our quantitative assessment relies on a suite of standard metrics: R-Precision (Top-1/2/3) to measure text-motion matching accuracy; Fréchet Inception Distance (FID) and MM-Dist to evaluate distributional similarity to real motions; Multi-Modality to quantify the diversity of generated samples per prompt; and CLIP Score to assess the fine-grained alignment via cosine similarity in a shared embedding space.

## A.2 HUMANML3D ANNOTATIONS

To provide a clearer understanding of our annotation processing pipeline described in Section 3.1, we present several examples from the HumanML3D dataset. The original annotations are free-form, descriptive sentences that often contain multiple sub-actions, stylistic modifiers, and narrative context, making them unsuitable for direct use as classification labels. Our two-stage process first utilizes a Large Language Model (LLM) to extract core action verbs and phrases. Subsequently, a clustering algorithm groups these extracted terms into a coherent set of semantic action classes, consolidating synonyms and variations (e.g., 'walks' and 'walking' into a single 'walk' class). Table 9 illustrates this transformation, showing the original text description alongside the final, structured multi-label action set used to train our Multi-modal Action Recognizer (MAR).

## A.3 IMPLEMENTATION DETAILS

Our experiments are conducted on two NVIDIA RTX 4090 GPUs. The multi-modal autoencoder (AE) comprises an encoder and a decoder, both based on the ResNet architecture. It first embeds the three modalities using 1D convolutional layers and then performs early fusion before feeding the input into a shared encoder. The encoder downsamples the motion sequence by a factor of 4, mapping the input features into a 512-dimensional latent space ($D = 512$). The masked autoregressive Transformer consists of a 6-layer Transformer model with a latent dimension of 1024. For the diffusion model, we utilize a multilayer perceptron (MLP) architecture. All models are trained using the AdamW optimizer with a learning rate of $2 \times 10^{-4}$. The multimodal action recognizer (MAR) is also a Transformer-based model with an embedding space dimension of 512, trained using a contrastive loss with a temperature parameter $\tau = 0.1$. During inference, the guidance scale $\gamma$ for semantic guidance is set to 1.0.

Table 9: Examples of the annotation transformation process on the HumanML3D dataset. Original free-form text descriptions are converted into a structured set of multi-label action classes. The processed labels are more atomic, consistent, and suitable for training a recognition model.

| Original Text Description | Processed Action Labels |
|---|---|
| bends to the right, picks up something, turns, then sets it down. | ['bend right', 'pick up', 'turn', 'put down'] |
| a person mimics playing the guitar. | ['play guitar'] |
| a person walking in a diagonal line. | ['walk diagonal'] |
| the person is swinging their hands out. | ['swing arms'] |
| a person stayed on the place and raised the left hand | ['raise left hand'] |
| a person walks forward, spins around on his right leg clockwise, then walks back into the direction he came from | ['walk forward', 'spin', 'walk backward'] |
| a man jogs in a very wide counterclockwise spiral, moving his arms. | ['jog in circle'] |
| a person walks backwards quickly | ['walk backward'] |
| a person shuffles to the side, then walks forward and nearly misses tripping. | ['shuffle', 'walk forward', 'trip'] |
| person picked up an object and moved it over | ['pick up', 'move object'] |
| a person crouches around and puts their hands to their ear to hear something. | ['crouch', 'listen'] |
| a person walks sideways to the right for a few steps and then walks sideways to the left for a further distance | ['walk sideways'] |

## B  SUPPLEMENTARY RESULTS

### B.1  ADDITIONAL QUALITATIVE RESULTS

More extensive qualitative results, including videos of the generated motions, are available for review at the following anonymous link: https://anonymous.4open.science/w/CoAMD-26D4/. To provide a more intuitive understanding of our method's capabilities, we present a series of qualitative comparisons in Figure 5. The figure visualizes motions generated by the unguided baseline (w/o MAR) and our full model with semantic guidance (w/ MAR) for a diverse range of challenging text prompts from the HumanML3D dataset. These examples highlight the significant impact of our MAR-based guidance on improving the semantic accuracy and expressiveness of the generated motions. While the unguided baseline often produces plausible but generic movements, our guided model consistently synthesizes motions that are more faithful to the fine-grained details in the text. For instance, in Figure 5(a), our model accurately captures the elegant posture and arm movements characteristic of a "waltz dance step", whereas the baseline produces a simple, generic turning motion. Similarly, in prompt (h), our model successfully generates "exaggerated steps" and wide arm swings, capturing the specified style of walking. The baseline, in contrast, generates a standard walk. Furthermore, our guidance mechanism demonstrates a strong ability to interpret complex, multi-part actions and specific body part manipulations. In prompt (d), our model correctly generates the action of "lifts the left hand and glances at the watch", a nuanced interaction that the baseline fails to produce. In prompt (i), the model accurately portrays the emotion of sadness by having the figure "raise a hand to the face, and wipe away tears", showcasing its capacity for generating emotionally expressive movements. These visual results underscore the effectiveness of our semantic guidance in bridging the gap between high-level text descriptions and the low-level dynamics of realistic, detailed human motion.

### B.2  USER STUDIES

In addition to quantitative metrics, we conducted two user studies to evaluate our method's performance from the perspective of human perception, focusing on motion quality and semantic consistency. We recruited 20 participants, each of whom evaluated 15 sets of motions rendered as videos. Motion Preference. In this study, we aimed to assess the overall quality and naturalness of the generated motions. For each text prompt, participants were shown three anonymous videos side-by-side: one from our guided model (CoAMD), one from a strong baseline (MARDM), and the ground truth

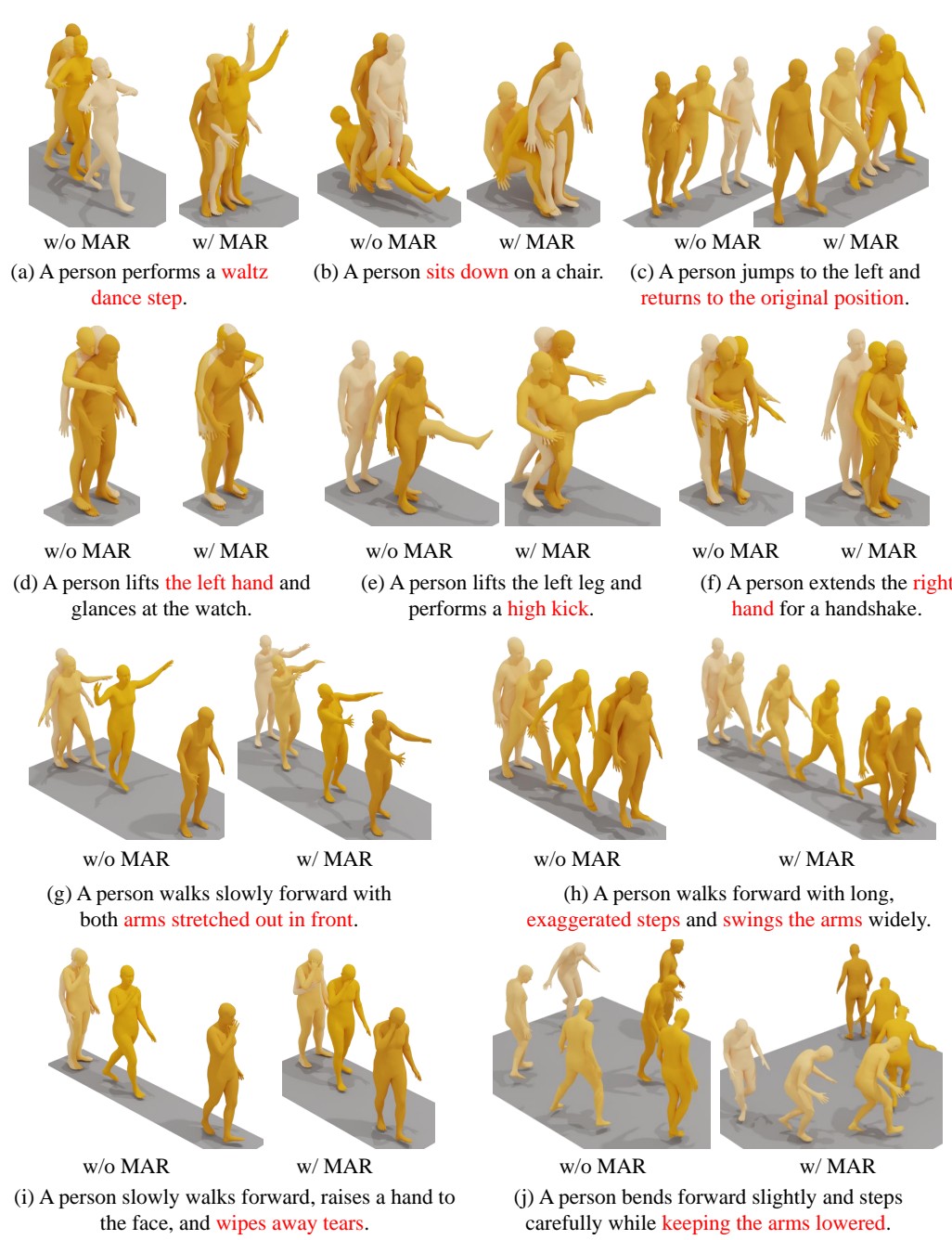

Figure 5: Qualitative comparison of generated motions on the HumanML3D dataset. For each text prompt, we show the motion generated by our unguided baseline (w/o MAR) and our full model with semantic guidance (w/ MAR). Our guided approach consistently produces motions that are more semantically aligned with the fine-grained details of the text prompts (highlighted in red).

motion (GT). They were asked to select the motion they preferred the most. As shown in Table 10(a), motions generated by our model were preferred in 45.2% of cases, significantly outperforming the baseline and demonstrating a quality that approaches that of real motion captures. Semantic Consistency. This study was designed to specifically evaluate the text-motion alignment. For each text prompt, participants were shown two videos: one from our unguided baseline (w/o MAR) and one from our full, guided model (w/ MAR). They were asked to answer the question: "Which motion

better aligns with the text description, or are they both equally good/bad?". The results, visualized in Figure 10(b), show a clear and consistent preference for our guided model. Across all votes, motions generated with our MAR guidance were favored 75.8% of the time, confirming that our semantic guidance mechanism provides a statistically significant improvement in generating motions that are faithful to the textual prompts.

Table 10: User study results. (a) Motion Preference (%): Comparison of our guided model (CoAMD) against a strong baseline (MARDM) and Ground Truth (GT). (b) Semantic Consistency (%): Preference for our guided model (w/ MAR) versus the unguided baseline (w/o MAR).

| (a) Motion Preference (%) | | | (b) Semantic Consistency (%) | | |
|---|---|---|---|---|---|
| CoAMD (ours) | MARDM | GT | w/ MAR | w/o MAR | Neither |
| 45.2 | 15.5 | 39.3 | 75.8 | 16.7 | 7.5 |

## C  MATHEMATICAL DERIVATIONS

### C.1  AUTOREGRESSIVE MOTION DIFFUSION WITH ODE SAMPLING

Our diffusion model is based on the continuous-time probability flow ODE formulation. The process transforms a simple prior distribution $\pi(\mathbf{z}_{m,1}) = \mathcal{N}(0, \mathbf{I})$ into a complex data distribution $p_0(\mathbf{z}_{m,0})$.

**Forward Process (Probability Flow ODE):** The forward process defines a trajectory from a data point $\mathbf{z}_{m,0}$ to a noise vector $\mathbf{z}_{m,1}$. We use a linear interpolation path, where the state at time $t \in [0, 1]$ is:

$$\mathbf{z}_{m,t} = (1 - t)\mathbf{z}_{m,0} + t\boldsymbol{\epsilon}, \quad \text{where } \boldsymbol{\epsilon} \sim \mathcal{N}(0, \mathbf{I}). \tag{9}$$

The probability density $p_t(\mathbf{z}_{m,t})$ of the variable $\mathbf{z}_{m,t}$ follows the continuity equation. The velocity field of this probability flow is given by the time derivative of the path:

$$\mathbf{v}(\mathbf{z}_{m,t}) = \frac{d\mathbf{z}_{m,t}}{dt} = \boldsymbol{\epsilon} - \mathbf{z}_{m,0}. \tag{10}$$

Our neural network $\mathbf{v}_\theta(\mathbf{h}, t)$ is trained to approximate this velocity field, conditioned on the context $\mathbf{h}$ from the MAR-Transformer. The training objective is to minimize the L2 loss:

$$\mathcal{L}_{\text{diff}} = \mathbb{E}_{\mathbf{z}_{m,0}, \boldsymbol{\epsilon}, t, \mathbf{h}} \|\mathbf{v}_\theta(\mathbf{h}, t) - (\boldsymbol{\epsilon} - \mathbf{z}_{m,0})\|^2. \tag{11}$$

**Reverse Process (Generative Sampling):** To generate a sample, we solve the reverse-time ordinary differential equation (ODE) from $t = 1$ to $t = 0$. The generative ODE is defined by the learned velocity field:

$$\frac{d\mathbf{z}_{m,t}}{dt} = \mathbf{v}_\theta(\mathbf{h}, t). \tag{12}$$

Starting with an initial sample from the prior distribution, $\mathbf{z}_{m,1} \sim \mathcal{N}(0, \mathbf{I})$, we can obtain the final data sample $\mathbf{z}_{m,0}$ by integrating this ODE:

$$\mathbf{z}_{m,0} = \mathbf{z}_{m,1} - \int_0^1 \mathbf{v}_\theta(\mathbf{h}, t) \, dt. \tag{13}$$

In practice, this integral is approximated numerically using a solver, such as the Euler method. For a discrete number of steps $N$, starting with $\mathbf{z}_{m,t_i}$ where $t_i = 1 - i/N$, the update rule is:

$$\mathbf{z}_{m,t_{i+1}} = \mathbf{z}_{m,t_i} - \frac{1}{N}\mathbf{v}_\theta(\mathbf{h}, t_i). \tag{14}$$

Iterating this process from $i = 0$ to $N - 1$ yields the final sample $\mathbf{z}_{m,0}$.

### C.2  DERIVATION OF THE SEMANTIC GUIDANCE GRADIENT

The semantic guidance mechanism modifies the reverse sampling process by incorporating the gradient of a semantic alignment score $\mathcal{S}$. This can be viewed as sampling from a modified probability

distribution that is a product of the original diffusion model's distribution and a guidance distribution.

**Modified Generative Process:** Let $p_t(\mathbf{z}')$ be the probability density at time $t$ induced by the unconditional diffusion model. We introduce a guidance distribution $p_{\text{guide}}(\mathbf{z}') \propto \exp(\mathcal{S}(\mathcal{D}(\mathbf{z}'), c))$ that assigns higher probability to samples with better text-motion alignment. The new, guided distribution $p_{\text{guided}}(\mathbf{z}')$ is proportional to their product:

$$p_{\text{guided}}(\mathbf{z}') \propto p_t(\mathbf{z}') \cdot p_{\text{guide}}(\mathbf{z}').$$

Taking the logarithm:

$$\log p_{\text{guided}}(\mathbf{z}') = \log p_t(\mathbf{z}') + \mathcal{S}(\mathcal{D}(\mathbf{z}'), c) + \text{const.} \tag{15}$$

The score function of a distribution is the gradient of its log-probability, $\nabla \log p$. Therefore, the score function of the guided distribution is:

$$\nabla_{\mathbf{z}'} \log p_{\text{guided}}(\mathbf{z}') = \nabla_{\mathbf{z}'} \log p_t(\mathbf{z}') + \nabla_{\mathbf{z}'} \mathcal{S}(\mathcal{D}(\mathbf{z}'), c). \tag{16}$$

In score-based diffusion models, the score function is related to the predicted noise or velocity. For the flow-matching ODE, the velocity field can be seen as a proxy for the score. Thus, we can modify the original velocity field $\mathbf{v}_\theta$ by adding a term proportional to the gradient of the alignment score. The guided velocity field $\mathbf{v}_{\text{guided}}$ becomes:

$$\mathbf{v}_{\text{guided}}(\mathbf{h}, t) = \mathbf{v}_\theta(\mathbf{h}, t) + \gamma \cdot \nabla_{\mathbf{z}'} \mathcal{S}(\mathcal{D}(\mathbf{z}'), c). \tag{17}$$

where $\gamma$ is the guidance scale.

**Chain Rule for Gradient Computation:** The gradient term $\mathbf{g} = \nabla_{\mathbf{z}'} \mathcal{S}(\mathcal{D}(\mathbf{z}'), c)$ is computed using the chain rule. Let $\hat{\mathbf{X}} = \mathcal{D}(\mathbf{z}')$. The score $\mathcal{S}$ is a composite function $\mathcal{S}(\hat{\mathbf{X}}(z'))$.

$$\mathbf{g} = \frac{d\mathcal{S}}{d\mathbf{z}'} = \frac{\partial \mathcal{S}}{\partial \hat{\mathbf{X}}} \frac{\partial \hat{\mathbf{X}}}{\partial \mathbf{z}'}. \tag{18}$$

Let's break down the first term, $\frac{\partial \mathcal{S}}{\partial \hat{\mathbf{X}}}$. The score is:

$$\mathcal{S}(\hat{\mathbf{X}}, c) = \sum_{k \in \{f, j, b, m\}} w_k \cdot \text{sim}(R_\phi^k(\hat{\mathbf{X}}), E_{\text{text}}(c)),$$

$$\mathcal{S}(\hat{\mathbf{X}}, c) = \sum_k w_k \frac{R_\phi^k(\hat{\mathbf{X}}) \cdot E_{\text{text}}(c)}{\|R_\phi^k(\hat{\mathbf{X}})\| \|E_{\text{text}}(c)\|}. \tag{19}$$

The gradient with respect to the motion $\hat{\mathbf{X}}$ is then:

$$\frac{\partial \mathcal{S}}{\partial \hat{\mathbf{X}}} = \sum_k w_k \cdot \nabla_{\hat{\mathbf{x}}} \left( \frac{R_\phi^k(\hat{\mathbf{X}}) \cdot E_{\text{text}}(c)}{\|R_\phi^k(\hat{\mathbf{X}})\| \|E_{\text{text}}(c)\|} \right). \tag{20}$$

This involves the derivative of the cosine similarity and the Jacobian of the MAR's embedding function $R_\phi^k$. The second term in Eq. 18, $\frac{\partial \hat{\mathbf{X}}}{\partial \mathbf{z}'}$, is the Jacobian of the decoder network $\mathcal{D}$. While these terms have complex analytical forms, automatic differentiation libraries compute their product $\mathbf{g}$ efficiently via a single backpropagation pass. This computed gradient is then normalized and applied as per Equations 7 and 8 in the main text to steer the ODE sampling process.

