# OpenReview forum: "Multi-Modal Action Recognizer Bridges Human Motion Generation and Understanding"
_ICLR.cc/2026/Conference — Submitted to ICLR 2026_

### Official Review · Reviewer_KW4t · 2025-10-22

**Soundness:** 3
**Presentation:** 3
**Contribution:** 2
**Rating:** 4
**Confidence:** 3

**Summary:**

This paper presents Coordinates-based Autoregressive Motion Diffusion (CoAMD), a unified framework that bridges human motion generation and skeleton-based action recognition. The framework leverages a multi-modal motion representation, decomposing absolute joint coordinates into joint, bone, and motion streams. During autoregressive generation, the MAR computes semantic alignment scores for partially generated motions, and the gradients of these scores steer the diffusion model’s sampling trajectory to enhance text-motion alignment. Extensive experiments on text-to-motion generation, text–motion retrieval, and motion editing demonstrate that CoAMD achieves state-of-the-art performance.

**Strengths:**

The paper unifies motion generation and action recognition in a single framework, named CoAMD, where a powerful action recognizer provides real-time semantic guidance during the diffusion process. Extensive experiments on 13 benchmarks demonstrate that CoAMD achieves state-of-the-art performance.

**Weaknesses:**

Previous research has established unified models for motion generation and understanding, such as LMM[1] and MotionLLM[2]. Although these prior works primarily rely on RGB videos or images as input for motion understanding, RGB video data is more easily accessible and widely applicable in real-world scenarios compared to joint-based motion data. This prevalence of RGB-based approaches somewhat diminishes the novelty of this work.
[1] Zhang M, Jin D, Gu C, et al. Large motion model for unified multi-modal motion generation[C]//European Conference on Computer Vision. Cham: Springer Nature Switzerland, 2024: 397-421.
[2] Chen L H, Lu S, Zeng A, et al. Motionllm: Understanding human behaviors from human motions and videos[J]. arXiv preprint arXiv:2405.20340, 2024.

**Questions:**

1. Why is there a discrepancy between the performance of MoMask reported in this paper on the HumanML3D and KIT datasets and the results presented in the original MoMask paper?
2. While this work introduces a model that unifies motion generation and action recognition, the comparative analysis in text-to-motion generation is limited to specialized, task-specific baselines, lacking comparisons with contemporary unified frameworks (e.g., LMM, MotionAnything).
[1] Zhang M, Jin D, Gu C, et al. Large motion model for unified multi-modal motion generation[C]//European Conference on Computer Vision. Cham: Springer Nature Switzerland, 2024: 397-421.
[2] Zhang Z, Wang Y, Mao W, et al. Motion anything: Any to motion generation[J]. arXiv preprint arXiv:2503.06955, 2025.

---

> ### Author Response · Authors · 2025-11-21
>
> We sincerely thank the reviewer for their positive assessment of our work. We are encouraged that the reviewer acknowledges our unified framework and the state-of-the-art performance demonstrated through extensive experiments. We have revised our manuscript to address the concerns and will provide a point-by-point response below.
>
> **Q1: The novelty is diminished by prior unified models (LMM, MotionLLM) that use RGB data, which is more prevalent.**
>
> **A1:** We thank the reviewer for raising this important point regarding the landscape of unified models. We have cited these important works in the **Related Work section of our revised manuscript**. We respectfully argue that while RGB-based unified models like MotionLLM are significant, our skeleton-based approach addresses a **fundamentally different problem domain with distinct challenges and applications**, and thus represents a novel and necessary contribution.
>
> *   **Different Domains & Applications**: RGB-based methods excel at understanding motion from "in-the-wild" videos but are susceptible to background clutter and appearance variations. In contrast, skeleton-based methods capture the **pure, underlying 3D kinematics of human motion**. This level of precision is critical for applications where the exact body structure and dynamics are paramount, such as character animation, virtual avatars, robotics, and biomechanical analysis.
>
> *   **Novelty in the Skeleton Domain**: Our work is the **first to bridge skeleton-based action recognition with motion generation**. While prior unified models exist, none have focused on this specific, challenging, and important modality pairing. Our key novelty lies in designing a strong, expert action recognizer (MAR) and using its gradients to iteratively guide the generation process, a mechanism tailored specifically to the structured nature of skeleton data.
>
> **Q2: Why is there a discrepancy in the performance of MoMask?**
>
> **A2:** We thank the reviewer for this critical question, which highlights a crucial detail of our experimental setup necessary for a fair comparison. The discrepancy arises because, as stated in the caption of Table 1 and Appendix A.1, **all models are assessed using absolute coordinates**. In contrast, many original papers (including MoMask) report results using the HumanML3D representation, which uses relative coordinates and includes additional redundant features (e.g., joint velocities).
>
> To ensure transparency and rigor, we undertook the considerable effort of re-evaluating every baseline method on this absolute coordinate system. The results are presented in the table below.
>
> **Table: Comparison of baseline performance on HumanML3D under different motion representations.**
>
> | Method | Representation      | R-Precision Top 1 | R-Precision Top 2 | R-Precision Top 3 |
> | :----- | :------------------ | :---------------- | :---------------- | :---------------- |
> | MARDM  | Original (Relative) | 0.500             | 0.695             | 0.795             |
> | MARDM  | Ours (Absolute)     | 0.486             | 0.680             | 0.780             |
> | MoMask | Original (Relative) | 0.521             | 0.713             | 0.807             |
> | MoMask | Ours (Absolute)     | 0.493             | 0.686             | 0.784             |
>
> As shown, when we use the original relative representation, our results closely match those reported in the original papers. Our goal was not only to ensure a fair comparison but also to provide a valuable reference for the community, as research on absolute coordinates is becoming increasingly prominent. This facilitates future work by establishing a clear and consistent benchmark. The numbers in our table are the direct results of this extensive re-evaluation. We have added further clarification on this in the revised paper.
>
> **Q3: The comparative analysis lacks comparisons with contemporary unified frameworks (e.g., LMM, MotionAnything).**
>
> **A3:** We thank you for providing these important references and have cited them **in our revised paper**. We agree that a direct comparison in Table 1 would be valuable. However, a core principle of our work is a rigorous and fair evaluation where all methods are benchmarked on the same **absolute coordinate** system.
>
> Unfortunately, this was not feasible for the suggested methods. The official source code for LMM does not support re-evaluation on the absolute coordinate system, and MotionAnything has not yet released its source code. As we could not ensure a fair comparison, we were unable to include them in Table 1 but have discussed their contributions in the related work section.
>
> ***
>
> We thank the reviewer again for their constructive feedback, which has helped us improve the positioning and clarity of our work. We hope our responses and the revisions in the manuscript have addressed the reviewer's concerns.

---

### Official Review · Reviewer_69f2 · 2025-10-28

**Soundness:** 3
**Presentation:** 3
**Contribution:** 2
**Rating:** 4
**Confidence:** 5

**Summary:**

This study proposes a Coordinates-based Autoregressive Motion Diffusion model (CoAMD), which achieves coarse-to-fine human motion synthesis guided by a Multi-modal Action Recognizer (MAR) at the semantic level. The model can be applied to four tasks: action recognition, text-to-motion generation, text–motion retrieval, and motion editing, and achieves state-of-the-art performance across 13 benchmarks.

**Strengths:**

1. Coarse-to-fine motion generation improves the quality of motion generation.
2. Excellent cross-task performance and strong generalization ability.

**Weaknesses:**

Major Issues:
1.The core idea of the paper lies in the mutual enhancement between generation and understanding tasks. However, this idea has already been adopted in several previous works [1][2], leading to insufficient novelty.
2.In the multimodal motion recognizer, compared with the approach that divides the human body into five parts and uses contrastive learning to align text and motion, what advantages does the proposed method offer? Is it actually a subset of such approaches? Furthermore, could the overlap among joint points, skeleton structures, and motion dynamics introduce interference in motion generation?
3.The masking strategy in COORDINATES-BASED AUTOREGRESSIVE MOTION DIFFUSION originates from MoMask, and has also been used in MMM, BAMM, and LaMP. What advantages does the proposed method have compared to these approaches?
4.The comparison methods in Table 1 are incomplete. The paper focuses on the mutual enhancement and fine-grained modeling between behavior generation and understanding, yet lacks comparison with related works such as StableMoFusion, LaMP, KinMo, MG-MotionLLM, UniMotion, and Motion-Agent.
5.There are concerns about the experimental results: in Table 2, the FID of MoMask is 0.372 & 0.204 in the original paper, but 0.523 in this paper. This discrepancy raises doubts about the correctness and reliability of the experimental results. If the MoMask results are accurate, the proposed CoAMD method would not achieve the best performance on the KIT-ML dataset.
6.In the section on limitations and future work, the paper fails to address the critical aspect of physical plausibility. Can the current method ensure the physical rationality of generated motions?

Minor Issues:

1.In the introduction, there are too many commas, making the sentences disjointed. Additionally, please include proper citations when discussing application scenarios.
2.In Figure 2, the encoder icon is incorrect; please clarify what “h” represents.
3.Highlight or bold the best experimental results for better readability.

References:
[1] Cycle-Consistent Learning for Joint Layout-to-Image Generation and Object Detection
[2] Dual Reciprocal Learning of Language-based Human Motion Understanding and Generation

**Questions:**

Major Issues:
1.The core idea of the paper lies in the mutual enhancement between generation and understanding tasks. However, this idea has already been adopted in several previous works [1][2], leading to insufficient novelty.
2.In the multimodal motion recognizer, compared with the approach that divides the human body into five parts and uses contrastive learning to align text and motion, what advantages does the proposed method offer? Is it actually a subset of such approaches? Furthermore, could the overlap among joint points, skeleton structures, and motion dynamics introduce interference in motion generation?
3.The masking strategy in COORDINATES-BASED AUTOREGRESSIVE MOTION DIFFUSION originates from MoMask, and has also been used in MMM, BAMM, and LaMP. What advantages does the proposed method have compared to these approaches?
4.The comparison methods in Table 1 are incomplete. The paper focuses on the mutual enhancement and fine-grained modeling between behavior generation and understanding, yet lacks comparison with related works such as StableMoFusion, LaMP, KinMo, MG-MotionLLM, UniMotion, and Motion-Agent.
5.There are concerns about the experimental results: in Table 2, the FID of MoMask is 0.372 & 0.204 in the original paper, but 0.523 in this paper. This discrepancy raises doubts about the correctness and reliability of the experimental results. If the MoMask results are accurate, the proposed CoAMD method would not achieve the best performance on the KIT-ML dataset.
6.In the section on limitations and future work, the paper fails to address the critical aspect of physical plausibility. Can the current method ensure the physical rationality of generated motions?

Minor Issues:

1.In the introduction, there are too many commas, making the sentences disjointed. Additionally, please include proper citations when discussing application scenarios.
2.In Figure 2, the encoder icon is incorrect; please clarify what “h” represents.
3.Highlight or bold the best experimental results for better readability.

References:
[1] Cycle-Consistent Learning for Joint Layout-to-Image Generation and Object Detection
[2] Dual Reciprocal Learning of Language-based Human Motion Understanding and Generation

---

> ### Author Response · Authors · 2025-11-23
>
> We sincerely thank the reviewer for their thorough and expert feedback. We are encouraged that the reviewer recognizes the generation strategy and our model's excellent cross-task performance. We have made significant revisions to the manuscript to address the concerns raised and will provide a point-by-point response below.
>
> **Major Issues**
>
> **Q1: The core idea of mutual enhancement between generation and understanding.**
>
> **A1:** We thank you for providing these valuable references and have cited them in our revised manuscript. As we state in the *Unified Human Motion Generation and Understanding* paragraph of the Related Work section, while related: "these works treat motion understanding merely as motion-text retrieval and motion captioning, and no existing study has yet bridged skeleton-based human action recognition with motion generation, a gap our work seeks to fill."
>
> In our framework, we employ an explicit **action recognizer**, for which absolute coordinates serve as the natural input representation. To this end, we introduce an action recognition benchmark that provides a concrete platform for evaluating action recognition and for addressing this gap.
>
> Furthermore, unlike prior approaches that focus on joint training or cycle-consistent training, our method features a strong, pre-trained action recognizer (MAR) that acts as an active, gradient-based semantic guide. At each step of our autoregressive process, MAR supplies real-time corrective feedback that steers the sampling trajectory toward semantically more accurate outputs. This tight and iterative integration of an expert recognizer into the generation loop constitutes a novel mechanism.
>
> **Q2: Advantages of the multi-modal recognizer (MAR) and potential interference.**
>
> **A2:** In the field of action recognition, segmenting the body into five parts is indeed a common technique to improve feature extraction and accuracy. However, in our framework, MAR serves a dual purpose: recognition and guiding generation. For the generation task, we align the motion and text representations as a **holistic entity**. This ensures the **overall coherence and fluidity** of the generated motion, preventing artifacts that might arise from optimizing body parts independently.
>
> The Bone (B) and Motion (M) modalities are introduced to provide richer supervisory signals to the primary Joint (J) modality. They help the model better understand the underlying structure and dynamics, ensuring the generated absolute coordinates align more faithfully with the semantic guidance. As our ablation in **Table 7** confirms, they provide synergy, not interference.
>
> **Q3: The advantage of our method given the masking strategy is from prior work.**
>
> **A3:** We acknowledge that the masked autoregressive framework itself is a well-established and effective technique. Our contribution is not in the masking strategy, but in **what we do within that framework**.
>
> Our key advantage is the  **tight integration of iterative semantic guidance at each generation step**. Specifically, at each step of the generation process, we use the gradient of the MAR's alignment score to refine the latent prediction. This mechanism ensures that the generated motion is continuously steered to contain more accurate semantic information, a contribution distinct from prior works.

---

> > ### Author Response · Authors · 2025-11-23
> >
> > **Q4: Discrepancy in experimental results for MoMask.**
> >
> > **A4:** We thank the reviewer for this critical question, which highlights a crucial detail of our experimental setup necessary for a fair comparison. The discrepancy arises because, as stated in the caption of Table 1 and Appendix A.1, **all models are assessed using absolute coordinates**. In contrast, many original papers (including MoMask) report results using the HumanML3D representation, which uses relative coordinates and includes additional redundant features (e.g., joint velocities).
> >
> > To ensure transparency and rigor, we undertook the considerable effort of  re-evaluating every baseline method on this absolute coordinate system. The results are presented in the table below.
> >
> > **Table : Comparison of baseline performance on HumanML3D under different motion representations.**
> >
> > | Method | Representation      | R-Precision Top 1 | R-Precision Top 2 | R-Precision Top 3 |
> > | ------ | ------------------- | ----------------- | ----------------- | ----------------- |
> > | MARDM  | Original (Relative) | 0.500             | 0.695             | 0.795             |
> > | MARDM  | Ours (Absolute)     | 0.486             | 0.680             | 0.780             |
> > | MoMask | Original (Relative) | 0.521             | 0.713             | 0.807             |
> > | MoMask | Ours (Absolute)     | 0.493             | 0.686             | 0.784             |
> >
> > As shown, when we use the original relative representation, our results closely match those reported in the original papers. Our goal was not only to ensure a fair comparison but also to provide a valuable reference for the community, as research on absolute coordinates is becoming increasingly prominent. This facilitates future work by establishing a clear and consistent benchmark. The numbers in our table are the direct results of this extensive re-evaluation.
> >
> >
> >
> > **Q5: The comparison methods in Table 1 are incomplete.**
> >
> > **A5:** We thank the reviewer for this crucial suggestion. We agree that a comprehensive comparison with recent state-of-the-art methods is essential for contextualizing our work.
> >
> > In our **revised manuscript, we have now cited all the works** you mentioned. Furthermore, we have expanded **Table 1** to include a new comparison with **StableMoFusion**, which we successfully re-evaluated on our standardized **absolute coordinate** system to ensure a fair benchmark. The key results are shown below:
> >
> > **Table: StableMoFusion evaluated in absolute coordinates.**
> >
> > | Method         | R-Precision Top 1 ↑ | FID ↓ | MM-Dist ↓ | MModality ↑ | CLIP-score ↑ |
> > | -------------- | ------------------- | ----- | --------- | ----------- | ------------ |
> > | StableMoFusion | 0.514               | 0.111 | 3.023     | 1.811       | 0.666        |
> > | CoAMD (ours)   | 0.519               | 0.065 | 2.959     | 2.014       | 0.674        |
> >
> >
> >
> > Regarding the other methods you suggested, we were unfortunately unable to include them in our quantitative comparison for the following reasons, which prevent a rigorous and fair re-evaluation on absolute coordinates:
> >
> > - KinMo, MG-MotionLLM, and UniMotion could not be included as their official source code has not been made publicly available.
> > - For LaMP, the official repository does not provide the pre-trained model weights required for evaluation.
> > - For Motion-Agent, the officially provided pre-trained weights produced results that differed from those reported in their original paper.
> >
> >
> >
> > **Q6: The paper fails to address physical plausibility in the limitations.**
> >
> > **A6:** We appreciate the reviewer for pointing out this important omission. While our model learns realistic dynamics implicitly by training on a large corpus of real human motion, it does not have explicit physical constraints. We have now added a discussion on this to the **Limitations and Future Work section of our revised manuscript**. We state that integrating physics-based simulators or loss functions to explicitly enforce physical plausibility is a promising and important direction for future research.
> >
> >
> >
> >
> > ### **Minor Issues**
> >
> > 1.  **Writing Style & Citations**: Thank you for the suggestion. We have revised the introduction and added the necessary citations in the revised manuscript.
> > 2.  **Figure 2**: We thank you for pointing this out. We have corrected the figure. The variable 'h' represents the contextual embedding from the transformer, as defined in **Equation (3)**.
> > 3.  **Highlighting Results**: This has been done. All tables in the revised manuscript now have the best results **bolded**.
> >
> > ***
> >
> > We thank the reviewer again for their highly valuable and constructive feedback, which has significantly strengthened our paper. We hope our detailed responses and the corresponding revisions have addressed all concerns.

---

### Official Review · Reviewer_FTNv · 2025-10-31

**Soundness:** 3
**Presentation:** 2
**Contribution:** 2
**Rating:** 4
**Confidence:** 3

**Summary:**

The paper argues that motion generation and recognition both build alignment between text and motion and can complement each other. Because of it, the paper proposed CoAMD, a motion generation model with a recognition model that provides guidance. Experiments were done on multiple datasets and different tasks to show the effectiveness and versatility of the proposed method.

**Strengths:**

The paper provides a new perspective of viewing action recognition and motion generation, and it builds a novel framework to combine models for the two tasks and reaches better performances.

**Weaknesses:**

1. The performance of baseline models in Table 1 is worse than that reported in the original papers. For example, in MoMask, the R-Presicion was 0.521, 0.713, 0.807 but what you listed was 0.493, 0.686, 0.784. I wonder where exactly these numbers come from?
2. You used LLM to extract core action verbs. Can you elaborate on the necessity of this step? An LLM is costly for such a task.
3. From the metrics you provided, the improvement of MAR seems to be very limited compared with your model without MAR.
4. Minor issues: Equation 5 is inconsistent with Equation 11 in the appendix. The shape of the encoder in Figure 2 is inverted — it goes from narrow to wide, which is counterintuitive.

**Questions:**

Please address the questions raised in the weakness section.

---

> ### Author Response · Authors · 2025-11-21
>
> We sincerely thank the reviewer for their detailed and insightful feedback. We are encouraged that the reviewer acknowledges our new perspective on unifying action recognition and motion generation and the novelty of our framework. We have revised our manuscript to address the concerns and will address each point below.
>
> **Q1: On the performance of baseline models in Table 1.**
>
> **A1:** We thank the reviewer for this critical question, which highlights a crucial detail of our experimental setup necessary for a fair comparison. The discrepancy arises because, as stated in the caption of Table 1 and Appendix A.1, **all models are assessed using absolute coordinates**. In contrast, many original papers (including MoMask) report results using the HumanML3D representation, which uses relative coordinates and includes additional redundant features (e.g., joint velocities).
>
> To ensure transparency and rigor, we undertook the considerable effort of  re-evaluating every baseline method on this absolute coordinate system. The results are presented in the table below.
>
> **Table: Comparison of baseline performance on HumanML3D under different motion representations.**
>
> | Method | Representation      | R-Precision Top 1 | R-Precision Top 2 | R-Precision Top 3 |
> | ------ | ------------------- | ----------------- | ----------------- | ----------------- |
> | MARDM  | Original (Relative) | 0.500             | 0.695             | 0.795             |
> | MARDM  | Ours (Absolute)     | 0.486             | 0.680             | 0.780             |
> | MoMask | Original (Relative) | 0.521             | 0.713             | 0.807             |
> | MoMask | Ours (Absolute)     | 0.493             | 0.686             | 0.784             |
>
> As shown, when we use the original relative representation, our results closely match those reported in the original papers. Our goal was not only to ensure a fair comparison but also to provide a valuable reference for the community, as research on absolute coordinates is becoming increasingly prominent. This facilitates future work by establishing a clear and consistent benchmark. The numbers in our table are the direct results of this extensive re-evaluation. We have added further clarification on this in the revised paper.
>
> **Q2: On the necessity of using an LLM to extract verbs.**
>
> **A2:** We appreciate the reviewer's concern about the cost and necessity of using an LLM. This step is essential for the **action recognition** component of our unified framework. The original annotations in datasets like HumanML3D are not simple labels but **free-form, descriptive sentences with multiple sub-actions, stylistic modifiers, and narrative context** (as described in Appendix A.2).
>
> - **Necessity**: Simpler, rule-based methods (e.g., Part-of-Speech tagging and keyword extraction) are insufficient to handle this complexity. For example, for a description like *"a person who is running, stops, bends over and looks down while taking small steps, then resumes running"*, the LLM's advanced natural language understanding is required to parse this complex structure and accurately distill it into a coherent set of atomic action labels: ["run", "stop", "bend", "look down", "step"].
> - **Cost**: We want to clarify that the LLM is used only as a **one-time, offline preprocessing tool** to create our new benchmark annotations. It is not part of the model's training or inference pipeline. This one-off cost is manageable and standard for dataset creation and allows us to establish a high-quality, multi-label benchmark that benefits all subsequent research.
>
>
>
> **Q3: On the seemingly limited improvement from MAR.**
>
> **A3:** We appreciate this observation and would like to highlight that MAR's role is multifaceted and its benefits extend beyond a single metric.
>
> 1.  **Versatility**: As demonstrated in our paper (Tables 3-5), MAR is not just a guidance module for generation but also a powerful, state-of-the-art model for **action recognition and retrieval**, showcasing the versatility of our unified approach.
> 2.  **Qualitative Improvement**: For the generation task, it is crucial to also assess the visual quality. The qualitative results in Figure 4 and Figure 5 visually demonstrate MAR's impact. These visualizations clearly show that MAR's guidance is critical for generating nuanced and semantically accurate motions that the unguided model fails to produce.  For more qualitative visualizations, please refer to our anonymous project page: 🔗[**link**](https://anonymous.4open.science/w/CoAMD-26D4/).
>
> **Q4: On minor issues in Equation 5 and Figure 2.**
>
> **A4:** We sincerely thank you for catching these issues. We have corrected Equation 5 and Figure 2 in the **revised manuscript**.
>
> ***
>
> We thank the reviewer again for their constructive feedback, which has helped us improve the clarity and rigor of our paper. We hope our responses and the revisions in the manuscript have addressed the reviewer's concerns.

---

### Official Review · Reviewer_pGn3 · 2025-11-08

**Soundness:** 3
**Presentation:** 3
**Contribution:** 3
**Rating:** 6
**Confidence:** 3

**Summary:**

This paper proposes a unified framework called CoAMD that bridges skeleton-based action recognition and text-to-motion generation by leveraging a Multi-modal Action Recognizer (MAR) as an active semantic guide during motion synthesis. The method introduces a multi-modal motion representation using absolute joint coordinates decomposed into joint positions, bone vectors, and motion dynamics, which enhances both recognition and generation. CoAMD generates motion in a coarse-to-fine, masked autoregressive manner using a diffusion model conditioned on text and contextual information from a transformer, while at each generation step, gradients from the MAR—trained via contrastive learning on both fine-grained motion–text retrieval and high-level action classification—are used to steer the latent motion toward better semantic alignment. Extensive experiments across 13 benchmarks demonstrate state-of-the-art performance in skeleton-based action recognition, text-to-motion generation, motion–text retrieval, and motion editing, with ablation studies confirming the importance of both the multi-modal representation and iterative semantic guidance.

**Strengths:**

* It successfully bridges two traditionally separate tasks—skeleton-based action recognition and text-to-motion generation—within a single, coherent architecture, demonstrating that semantic understanding and motion synthesis are mutually reinforcing.
* Rather than treating the action recognizer as a passive evaluation tool, the paper uniquely employs the Multi-modal Action Recognizer (MAR) as an active, gradient-based semantic guide during the diffusion sampling process. This enables real-time correction and significantly improves text-motion alignment.
* By decomposing absolute skeleton coordinates into joint positions, bone vectors, and motion dynamics, the model captures richer spatio-temporal and kinematic information. This representation consistently boosts performance in both recognition and generation tasks.

**Weaknesses:**

* The proposed method is designed for single-agent motion and does not handle multi-person or interactive scenarios. Real-world applications often involve multiple agents with social or physical interactions, which this model cannot address.
* The iterative semantic guidance mechanism introduces additional computational cost during inference. At each autoregressive step, the model must decode motion, compute the semantic score via MAR, and backpropagate its gradient, slowing down generation compared to unguided or non-iterative approaches.

**Questions:**

The performance of the model largely depends on the clarity and structure of the input text prompts. Although MAR is trained on diverse HumanML3D descriptions, highly abstract, ambiguous, or poorly phrased instructions may still result in suboptimal motion synthesis. How does the model perform when provided with vague, abstract, or grammatically incorrect textual descriptions?

---

> ### Author Response · Authors · 2025-11-21
>
> We sincerely thank the reviewer for their detailed and highly positive review. We are particularly encouraged by their accurate and insightful summary of our work's strengths, including the novelty of bridging recognition and generation, the use of MAR as an active semantic guide, and the effectiveness of our multi-modal representation. We will address the reviewer's question below.
>
> **Q: How does the model perform with abstract textual descriptions?**
>
> **A:** This is an excellent and practical question. We acknowledge that handling highly abstract, ambiguous, or poorly phrased instructions remains a challenge that can lead to suboptimal motion synthesis, not just for our model but for the broader text-to-motion field.
>
> However, we find that our iterative semantic guidance provides a distinct advantage in interpreting and realizing such prompts. To demonstrate this, we invite the reviewer to view our qualitative results at our anonymous project page:🔗 [**link**](https://anonymous.4open.science/w/CoAMD-26D4/).
>
> Our model successfully generates nuanced motions for complex prompts. For example, it can produce faithful results for prompts like:
>
> *   *"a man stumbles stepping out to his left before returning to a standing position."*
> *   *"a person shimmies then starts exercising by lifting up their knees."*
> *   *"a person holds their left arm out on something to support them while sticking their left leg up to balance."*
>
> These examples, available for viewing at the link above, show how MAR's guidance helps capture detailed interactions and sequences that unguided models would likely miss. In summary, our model exhibits strong robustness to complex and grammatically imperfect prompts by capturing the core semantic intent, a strength we attribute to our iterative guidance mechanism.
>
> We thank the reviewer again for their insightful feedback and encouraging assessment of our work. We hope our responses have addressed the points raised.

---

### Author Response · Authors · 2025-12-01
**Global Response**

Dear Area Chair,

We sincerely thank all reviewers for their comments and constructive suggestions. To facilitate the decision-making process, we summarize the key feedback and our major updates below.

**Reviewer Feedback and Our Revisions**

- **Rigorous Re-evaluation on Absolute Coordinates**

We address the concerns raised by several reviewers regarding the performance discrepancies of baseline models compared to their original papers. We clarified that this stems from a crucial difference in evaluation protocols: our work evaluates all models on absolute coordinates, a more fundamental representation, whereas prior works often use relative coordinates. To ensure transparency and rigor, **we undertook the considerable effort of re-evaluating every baseline method on this absolute coordinate system, establishing a clear and consistent benchmark for future researchers.** The new results confirm that when using relative coordinates, our reproduction matches original papers, but under the rigorous absolute coordinate benchmark, **CoAMD achieves State-of-the-Art performance**, nullifying the concerns about baseline performance.

- **Comparisons with Unified Frameworks**

In response to reviewers' suggestions to include other unified models, we have expanded our comparisons and discussions. First, we successfully re-evaluated StableMoFusion on our standardized absolute coordinate system and added it to Table 1, where **CoAMD demonstrates superior performance in R-Precision, FID, and MM-Dist**. Second, regarding other suggested works like LMM and MotionAnything, we have cited them in the revised Related Work section but excluded them from quantitative tables because their official source codes or weights are either unavailable or do not support fair absolute coordinate evaluation. Third, we clarified that unlike RGB-based methods, **our work focuses on the distinct challenges of skeleton-based modeling**, which is critical for precise animation and robotics.

- **Effectiveness of Semantic Guidance and Robustness**

We addressed questions regarding the robustness and cost of our Multi-modal Action Recognizer (MAR). First, regarding robustness, we demonstrated that our iterative semantic guidance allows the model to handle abstract or grammatically imperfect prompts effectively, supported by qualitative video results on our anonymous project 🔗[**link**](https://anonymous.4open.science/w/CoAMD-26D4/). Second, regarding cost, we clarified that using an LLM to extract action verbs is a **one-time, offline preprocessing step essential for creating high-quality benchmarks**, imposing no runtime overhead. Third, we highlighted that **MAR serves as an active, gradient-based guide** that reduces FID and enables the generation of nuanced semantic details.



- **Limitations and Future Work**

We appreciated the suggestion regarding physical plausibility. We have updated the manuscript to explicitly discuss that while our model learns implicit dynamics from data, ensuring strict physical plausibility via physics-based constraints or simulators is a promising and important direction for future research.

**Our Core Contributions**

- **Unified Framework:** We introduce the first framework to bridge skeleton-based action recognition and motion generation, establishing a bidirectional connection between human motion and language semantics.
- **Active Semantic Guidance:** We propose CoAMD, a novel architecture that synergizes skeleton-based motion generation with a multi-modal action recognizer (MAR) to provide active, gradient-based guidance to the diffusion model.
- **SOTA Performance:** Our approach achieves state-of-the-art performance across 13 benchmarks on diverse tasks, rigorously validated through our comprehensive re-evaluation using absolute coordinates.

We hope these clarifications and revisions address the reviewers' concerns. We thank the reviewers for their efforts and kindly request the support of the Area Chair.

---

### Author Response · Authors · 2025-12-04

Dear Area Chair,

We respectfully request your attention to our submission and the accompanying rebuttal. While all reviewers acknowledged the novelty and effectiveness of our unified framework, they did not participate in the discussion in a timely manner. Consequently, the borderline scores do not reflect fundamental flaws, but rather specific misunderstandings that we have comprehensively resolved:

The Misunderstanding (Baselines): Three reviewers (FTNv, 69f2, KW4t) questioned discrepancies in baseline performance (e.g., MoMask).

Our Clarification: We performed **a rigorous re-evaluation of all baselines under the Absolute Coordinate setting**. This not only proved our method is State-of-the-Art but also **establishes a standardized benchmark for future research**. This directly nullifies the primary basis for their negative scores regarding performance.

We thank the reviewers for their efforts and kindly request the support of the Area Chair.

Best regards,

The Authors

---

### Meta-Review · Area_Chair_J5sv · 2026-01-03

**Summary:**

The main concern is that the paper’s core idea, mutual enhancement between motion generation and motion understanding, is not sufficiently novel, as similar bidirectional/reciprocal learning ideas have appeared in prior works. In this paper, they have used more advanced motion generation and action recognition networks/frameworks, but they are more like design choices instead of a new principle, and those networks are not new as well. It is a bit of an overclaim to say that the coordinates-based autoregressive motion diffusion model and the multi-modality representation are technical contributions, since they are not new but borrowed from existing works. Also, it is not fully convincing to call them multi-modal, since they remain different parameterizations/streams of the same 3D skeleton motion rather than distinct sensing modalities. The method introduces extra inference-time compute due to per-step gradient-based guidance, but the real cost of this inference-time gradient updating is not clearly quantified.

**Reviewer Concerns:**

Addressed: 1) Baseline discrepancy: the authors clarified the evaluation setting (absolute-coordinate representation) and explained why the re-evaluated baseline numbers differ from the originally reported results, though it is not clear what the rationale is for this choice. 2) LLM usage cost: The authors clarified that the LLM is used offline for annotation, not at training or inference time. 3) Physical plausibility: Explicitly acknowledged and added to the limitations/future work discussion.

Not fully addressed: 1) Comparison with other works: while authors cite additional work and add one extra comparison, reviewers specifically noted missing comparisons to several relevant unified frameworks; the rebuttal indicates some comparisons were not feasible under their evaluation setup, leaving the empirical positioning incomplete. 2) Inference-time cost is still not convincingly resolved; the authors didn't respond to this concern. 3) A reviewer noted the quantitative improvement from MAR seems limited, and the authors argued on versatility and qualitative examples rather than resolving the concern with stronger ablations or clearer attribution.

**Reviewer Scores:**

All four reviewers showed no indication of changing their scores. However, since Reviewer pGn3 questioned the inference time while the authors didn't provide a response, it is likely that Reviewer pGn3 might decrease the rating. Given the concerns that remain (summarized above), it is not expected that other reviewers will raise their scores.

---

### Decision · Program_Chairs · 2026-01-26

Reject